# Pathological Nodal Staging Score for Gastric Signet Ring Cell Carcinoma: A Clinical Tool of Adequate Nodal Staging

**DOI:** 10.3390/diagnostics12102289

**Published:** 2022-09-22

**Authors:** Chaoran Yu, Zhiyuan Zhou, Bin Liu, Danhua Yao, Yuhua Huang, Pengfei Wang, Yousheng Li

**Affiliations:** Department of General Surgery, Shanghai Ninth People’s Hospital, Shanghai Jiao Tong University School of Medicine, Shanghai 200025, China

**Keywords:** gastric signet ring cell carcinoma, beta-bimonial distribution, nodal staging score, prognostic analysis

## Abstract

**Background:** Gastric signet ring cell carcinoma (GSRCC) is a subset of gastric cancer with distinct histological and inconsistent prognosis outcome. Currently, the association between the adequate regional lymph node and proper nodal staging in GSRCC is rarely noticed. **Materials and methods:** Clinical data of GSRCC were retrieved from the Surveillance, Epidemiology, and End Results database. Beta-binomial distribution model was employed for the estimation of the probability of missing nodal disease, followed by the development of a nodal staging score (NSS). **Results:** A total of 561 GSRCC patients were included in this study, with 193 in lymph node-negative and 368 in lymph node-positive diagnoses. As the number of examined lymph nodes increased, the probability of missing nodal disease decreased rapidly, with T stage-specific curves. The probability of missing nodal disease in T4 was lower than that in T1. NSS calculation indicated that T1 stage patients commonly had NSS > 0.8. However, with the NSS of T2–T4 to reach 0.8, the number of examined lymph node was required to be larger than 12 in T2, 17 in T3 and 27 in T4. NSS ≥ 0.75 (quantile 75%) subgroup in T2–T4 subgroups tended to have better outcome; however, without significant prognostic value. **Conclusions:** NSS is served as a reliable and feasible tool in adequate nodal staging of GSRCC with statistical basis and provides further evidence for clinical decision making.

## 1. Introduction

Gastric signet ring cell carcinoma (GSRCC) is named for a signet-ring pathological resemblance found in a subset of gastric cancer (4–17%), featuring a large amount of mucin in tumor cells with the nucleus aside [1,2]. According to the latest histological classification of the World Health Organization, GSRCC is defined as poorly cohesive carcinoma type [1,3]. In addition, diffuse type or undifferentiated type is also commonly found in GSRCC. Given the fact that overall incidence of gastric cancer is downsizing, the remaining increased incidence of GSRCC continues to be challenging during the last thirty years [4,5,6]. Although the prognosis of GSRCC is unfavorable, inconsistent outcome was also reported [5,6,7,8,9,10]. A meta-analysis of 58 eligible studies reported that early GSRCC was associated with favorable outcome, while advanced GSRCC was more likely associated with a poorer outcome [8]. GSRCC served to be a favorable prognostic factor in stage I of GC, while as a poor factor in stage II/III [9]. In fact, 5-year survival rate of early GSRCC may reach 99.7% [2].

Noteworthy, early GSRCC demonstrates a relatively small proportion lymphatic metastasis within local region. A Korean Central Cancer Registry survey reports that regional lymph node metastasis of GSRCC occurred very less in mucosal stage of less than 3.9% patients, but with increased risk of poor outcome [11]. Standardized therapeutic management indicates that an extended D2 lymphadenectomy is necessary with at least 16 lymph nodes to be examined for adequate staging [12,13,14]. In a nation-level study in France with 899 GSRCC cases, a standard D2 or modified D2 lymphadenectomy was preferred [15]. However, others suggested extended lymphadenectomy may not benefit GSRCC, as D1/D1 plus lymphadenectomy could achieve a similar outcome [16].

Nonetheless, the jury is still out for the optimal number of dissected lymph nodes in GSRCC. A retrospective cohort study from Southern China demonstrates that a harvest of 7–15 perigastric lymph nodes could improve survival benefits [17]. In stage II gastric cancer, the minimal number of lymph nodes was increased to 30 [18]. However, investigation on the exact adequate number of dissected lymph node and proper nodal staging in GSRCC is rarely published.

Lymphatic recurrence of pathological N0 (pN0) gastric cancer has been increasingly noticed [19,20]. Five-year survival of pN0 GSRCC with early T stage was 82.8%, whereas in pN+ GSRCC, the survival was reduced to 68.2% [21]. This fact leads to the assumption that true positive lymph nodes with tumor metastasis may fail to be identified and remain hidden among the test negative lymph nodes cases. In order to quantificationally calculate the possibility of potential true positive lymph nodes and develop a tool for adequate nodal staging in clinical practice, we introduced a beta-binomial distribution-based model, nodal staging score (NSS). NSS has been employed for computing the probability of missing nodal disease in similar lymph node analysis in diverse cancer types [12,22,23,24].

In this study, NSS is used to estimate the possibility of a true positive lymph node in each individual and the possibility of cases with true negative lymphatic metastasis based on the Surveillance, Epidemiology, and End Results (SEER) database. This is the first investigation focusing on the NSS and its application in GSRCC.

## 2. Materials and Methods

All the clinical data of GSRCC, including TNM stage, sex, age, tumor size, bone metastasis, brain metastasis, liver metastasis, lung metastasis, tumor position, total examined lymph nodes, as well as positive lymph nodes, were retrieved from the SEER database (ID 16866-Nov2019). The SEER database program provides the largest and reliable cancer statistics based on the United States population over decades [25]. Only GSRCC patients with pathological confirmation of signet ring cell carcinoma (8490/3), complete TNM stage, follow-up outcome, and examined/positive lymph nodes were screened for analysis. A study consort diagram is presented in Figure 1.

This study is mainly based on the concerns that individuals with negative lymphatic metastasis could be incorrectly diagnosed. The method is based on previous algorithms initially described by Gönen et al. in 2009 and further modified by Robinson et al. in 2016 [23,24].

Calculation of the probability of missing true metastatic positive lymph node contained three pragmatic presumptions and three steps as following:

Presumption 1: No false positive in this scenario.

Presumption 2: All examined lymph nodes are theoretically exchangeable, without any consideration in sequence or position superiority.

Presumption 3: Sensitivity is computed based on the data of individual with positive lymph node metastasis with no difference between the true positive (TP) and false negative (FN) scenarios.

Step 1: Compute the probability of missing a metastatic positive lymph node as a function of all examined lymph nodes in each *T stage*, respectively.

The estimation of the probability of a false negative (FN) lymph node in the cases where one or more metastatic positive lymph nodes were identified among examined lymph nodes, individually. Next, compute the average probabilities following specified value of various examined lymph nodes as input data. To compute the probability of missing a positive lymph node for a given case, a *β*-binomial distribution model was introduced (Equation (1)):(1)$$\mathrm{Prob}(\mathrm{FN}|\mathrm{LN}\;\mathrm{examined},\;T\;\mathrm{stage})=\frac{\beta \left(\alpha_{T\;\mathrm{stage}},\beta_{T\;\mathrm{stage}}+\mathrm{LN}\;\mathrm{examined}\right)}{\beta \left(\alpha_{T\;\mathrm{stage}},\beta_{T\;\mathrm{stage}}\right)}$$
where *β*() represents the *β* function with maximum likelihood and *α_T stage_β_T stage_* represent given parameters relating to the probability intensity of lymph node status derived from input data. ‘*LN examined*’ indicates the total examined number of lymph nodes.

Step 2: Compute the prevalence of positive lymph node cases as a function of each *T stage*.

First, calculate the exact number of cases with *FN* at each given value of examined lymph node using Equation (2):(2)$$\#{\mathrm{FN}}_{\mathrm{LN}\;\mathrm{examined},\;T\;\mathrm{stage}\;}=\frac{\mathrm{Prob}\left(\mathrm{FN}|\mathrm{LN}\;\mathrm{examined},\;T\;\mathrm{stage}\right)*\left(\#\;{\mathrm{TP}}_{\mathrm{LN}\;\mathrm{examined},\;T\;\mathrm{stage}}\right)}{1-\mathrm{Prob}\left(\mathrm{FN}|\mathrm{LN}\;\mathrm{examined},\;T\;\mathrm{stage}\right)}$$
where ‘*LN examined*’ indicates the total examined number of lymph nodes and #*TP_LN examined, T stage_* indicates the exact number of cases with positive lymph nodes in given *T stage*.

Next, the calculation of prevalence is performed using Equation (3):(3)$${\mathrm{Prev}}_{T\;\mathrm{stage}}=\frac{\sum_{\mathrm{LN}\;\mathrm{examined}}\left(\#{\mathrm{TP}}_{\mathrm{LN}\;\mathrm{examined},\;T\;\mathrm{stage}}+\#{\mathrm{FN}}_{\mathrm{LN}\;\mathrm{examined},\;T\;\mathrm{stage}}\right)}{\sum_{\mathrm{LN}\;\mathrm{examined}}\left(\#{\mathrm{TP}}_{\mathrm{LN}\;\mathrm{examined},\;T\;\mathrm{stage}}+\#{\mathrm{TN}}_{\mathrm{LN}\;\mathrm{examined},\;T\;\mathrm{stage}}+\#{\mathrm{FN}}_{\mathrm{LN}\;\mathrm{examined},\;T\;\mathrm{stage}}\right)}$$

In each stage, the true prevalence is computed by summing up all cases with TP and FN, then divided by all patients.

Step 3: Compute the NSS based on the prevalence and the probability of missing a positive lymph node, using Equation (4):(4)$${\mathrm{NSS}}_{\mathrm{LN}\;\mathrm{examined},\;T\;\mathrm{stage}}=\frac{1-{\mathrm{Prev}}_{T\;\mathrm{stage}}}{1-{\mathrm{Prev}}_{T\;\mathrm{stage}}+{\mathrm{Prev}}_{T\;\mathrm{stage}}*\mathrm{Prob}(\mathrm{FN}|\mathrm{LN}\;\mathrm{examined},\;T\;\mathrm{stage})}$$

Confidential intervals (CIs) and survival analysis: to improve the precision of estimation, 1000 bootstraps were used to sample from the entire data with 95% CIs established. To further delineate the prognostic value of estimated NSS, four quantile groups of NSS, NSS < 0.25, 0.25 ≤ NSS < 0.5, 0.5 ≤ NSS < 0.75, and NSS ≥ 0.75, in each *T stage* have been divided.

## 3. Results

### 3.1. Clinical Characterization of the Included GSRCC Patients

A total of 561 GSRCC patients were included in this study based on the consort diagram (Figure 1, Table 1 and Appendix A). There were 193 patients in the lymph node metastatic-negative (LN−) group, with 368 in the lymph node metastatic-positive (LN+) groups. Among T stage, 98 out of 193 (50.8%) were T1 stage in LN− group, while 187 out of 368 (50.8%) were T4 stage in LN+ group. Significant positive lymph nodes were found in advanced T stage (Appendix A). Significant differences of several variables between the two groups were also found, including M stage (*p* = 0.001), tumor size (*p* < 0.001), and tumor position (*p* = 0.007). In total, 89 out of 193 (46.1%) patients in the LN− group presented with tumor size less than 3 cm, while 152 out of 368 (41.3%) patients in LN+ group presented with tumor size between 3–6 cm. However, no significant difference was identified in bone/brain/lung/liver metastasis.

### 3.2. Compute the Probability of Missing Potential Metastatic Lymph Node

By employing a *β*-binomial distribution model, the probability of missing potential metastatic lymph node, or false negative lymph node findings was computed based on the predefined input parameters *α_T stage_β_T stage_*, which are stratified by *T stage* (Table 2). Apparently, as the number of examined lymph nodes increased, the probability of missing nodal disease decreased rapidly (Figure 2). Additionally, the probability of missing nodal disease in advanced *T stage* was obviously lower than that in early *T stage*. The prevalence of nodal disease was adjusted and found to be higher than apparent prevalence given possible cases of false negative findings (Table 3).

### 3.3. Compution of the NSS

The computation of NSS indeed reflected the proportion of true negative lymph node of GSRCC patients among the lymph node negative group and the individual probability of true negative lymph node (Figure 3). Patients in T1 stage commonly had NSS > 0.8. As the examined number of lymph node increased, the NSS was close to 1.0. However, when the number of examined lymph node = 10, the NSS of T2 stage was less than 0.8, T3 less than 0.7, and T4 less than 0.6. Specifically, with the same level of NSS as 0.8, a minimal 12 examined lymph node of T2 was required, while in T3 and T4, the number was 17 and 27, respectively (Figure 3).

### 3.4. Prognostic Analysis of NSS-Based Group Classification

To further characterize the prognostic value of this tool, the overall survival curves of GSRCC patients divided by quantile NSS (NSS < 0.25, 0.25 ≤ NSS < 0.5, 0.5 ≤ NSS < 0.75, and NSS ≥ 0.75) were initially analyzed. The potential prognostic benefits of NSS had been noticed as NSS ≥ 0.75 subgroup in T2–T4 subgroups tended to have better outcome (Figure 4). However, no significant prognostic value was identified among NSS < 0.25, 0.25 ≤ NSS < 0.5, 0.5 ≤ NSS < 0.75, and NSS ≥ 0.75 groups in each *T stage* (Figure 4).

## 4. Discussion 

This study found out that the probability of failure to identify a false negative lymph node decreased as the more lymph nodes were to examined, and it varied in each *T stage*. By NSS estimation, T1 showed the highest probability comparing to T2–T4. To reach the same level of NSS, the minimal required number of lymph node was *T stage* specific. In fact, this study provided a quantificationally model to determine exact number of lymph nodes to be examined in advanced *T stage* for adequate lymph node staging.

The exact number of lymph nodes to be examined is one of the key issues in gastric cancer surgery. The 8th American Joint Committee on Cancer (AJCC) staging guideline has recommended a minimal requirement of 16 lymph nodes for improved lymph node staging, and 30 or more lymph nodes for accurate staging and prognosis prediction [14,26]. However, such investigation targeting GSRCC is rarely established. Some study has noticed a stage-specific lymph node retrieving may present diverse survival benefits in gastric cancer [18]. A study of 449 gastric cancer patients from Iran reported that the median total number of examined lymph nodes were 9 (ranging 0–55), with only 21.2% patients has adequate lymph nodes yielding (*n* ≥ 16) [27]. The National Comprehensive Cancer Network (NCCN) guideline suggests that a minimum 15 resecting lymph nodes is required, based on the evidence that 15 lymph nodes is the threshold number with a survival benefit [28].

Nonetheless, a Korean research paper opinioned that the optimal number of retrieved lymph nodes in early gastric cancer may not be 15 [29]. It reported that in the lymph node negative group, patients with more than 26 resected lymph nodes showed 90% 5-year survival rate and 75% 10-year survival, while patients with 15–25 yielding lymph nodes only showed 88% 5-year survival rate and 54% 10-year survival rate [29]. In addition, in lymph node positive group, the difference between more than 26 lymph nodes and 15–25 lymph nodes retrieval was more significant [29]. Interestingly, our study reported that the probability of missing metastatic lymph node in T1 was 11.77% (*n* = 15). When *n* = 25, the probability = 6.11%, and when *n* = 29, the probability reaches 4.99%. It could be more insightful and statistically valuable if the *n* = 29 was taken into consideration for threshold. Specifically, when the number of examined lymph node ≥29 in T1 stage, the probability of a false negative nodal disease will be <5%. Although initial prognostic analysis showed no significant difference between each subgroup of NSS quantile in T1–T4 stages, the potential prognostic benefits of NSS had been noticed as NSS ≥ 0.75 subgroup tended to have better outcome, which, however, required further larger sample to validate.

Other tools to assess regional lymph node in GSRCC have been released. Combined positron emission tomography/computed tomography (PET/CT) and contrast-enhanced CT (CECT) were employed for lymph node metastases evaluation retrospectively in 74 gastric cancer patients, of which 22% were GSRCC [30]. However, neither PET/CT nor CECT were able to detect positive lymph node in early gastric cancer. Moreover, lymph node metastasis in GSRCC turned out to be non-FDG-avid type [30]. Another study indicated that positive lymph node ratio could be an alternative indicator in GSRCC as well [31]. Moreover, endoscopic ultrasound (EUS) is mentioned as one of the gold standards for detecting positive regional lymph nodes [32]. However, accuracy of EUC is largely sensitive to the investigators and transducers’ techniques. Previous report indicated that the predictive value of this method could be as high as 94.1% in early gastric cancer and 92.6% in nodal negativity [33]. Nonetheless, a meta-analysis of 22 studies reported that the accuracy of EUS for nodal staging of gastric cancer significantly varied from 58.2% in N1, to 64.9% in N2 [34]. Meanwhile, magnetic resonance imaging (MRI) is found no superiority of nodal staging over other standard techniques, such as CT or EUS [35]. Therefore, NSS established in this study is served as a reliable, feasible, and alternative tool in the nodal staging of GSRCC.

The estimation of NSS is made based on a β-binomial model with three presumptions. First, no false positive case is taken into account in this analytic model. In the real world, false positive lymph node metastasis scenario is much rarer than the true positive lymph node being missed during pathological examination. Therefore, it is reasonable to achieve true negative and false negative data without too many uncertain variables.

Second, all the examined lymph nodes are exchangeable. Therefore, the establishment of NSS is purely based on probability rather than taking the stations of lymph nodes into consideration. This presumption is set up to facilitate the computation model. In reality, the lymph nodes resected from each station are not fully exchangeable. For example, in distal gastrectomy with lesion on gastric antrum, lymph nodes from No. 3, No. 5, No. 7, and No. 8 may tend to present higher positive results than No. 1, No. 4, and No. 6, due to lesion closeness [36]. Higher yielding lymph nodes from No. 1, No. 4, and No. 6 does not guarantee an equal level of tumor eradication from higher yielding lymph nodes from No. 3, No. 5, or No. 7.

Third, sensitivity is the same between true positive and false negative patients. In the real world, it is usually assumed that those positive cases that are pathologically recognized are easier than those false negative ones hiding among all negative cases, which indicates potential difference of sensitivity derived from true positive and false negative groups.

The limitation of this study is the relatively small sample size included. A total of 561 GSRCC patients were included in this study, with a relatively small sample size in each T stage, which may exert noticeable confounding bias on prognostic values of NSS subset groups. Perceivably, further research with larger sample sizes could improve the predictive power of NSS established in this study.

## 5. Conclusions

NSS is served as a reliable and feasible tool in adequate nodal staging of GSRCC with statistical basis and provides quantificational evidence for clinical decision making.

## Figures and Tables

**Figure 1 diagnostics-12-02289-f001:**
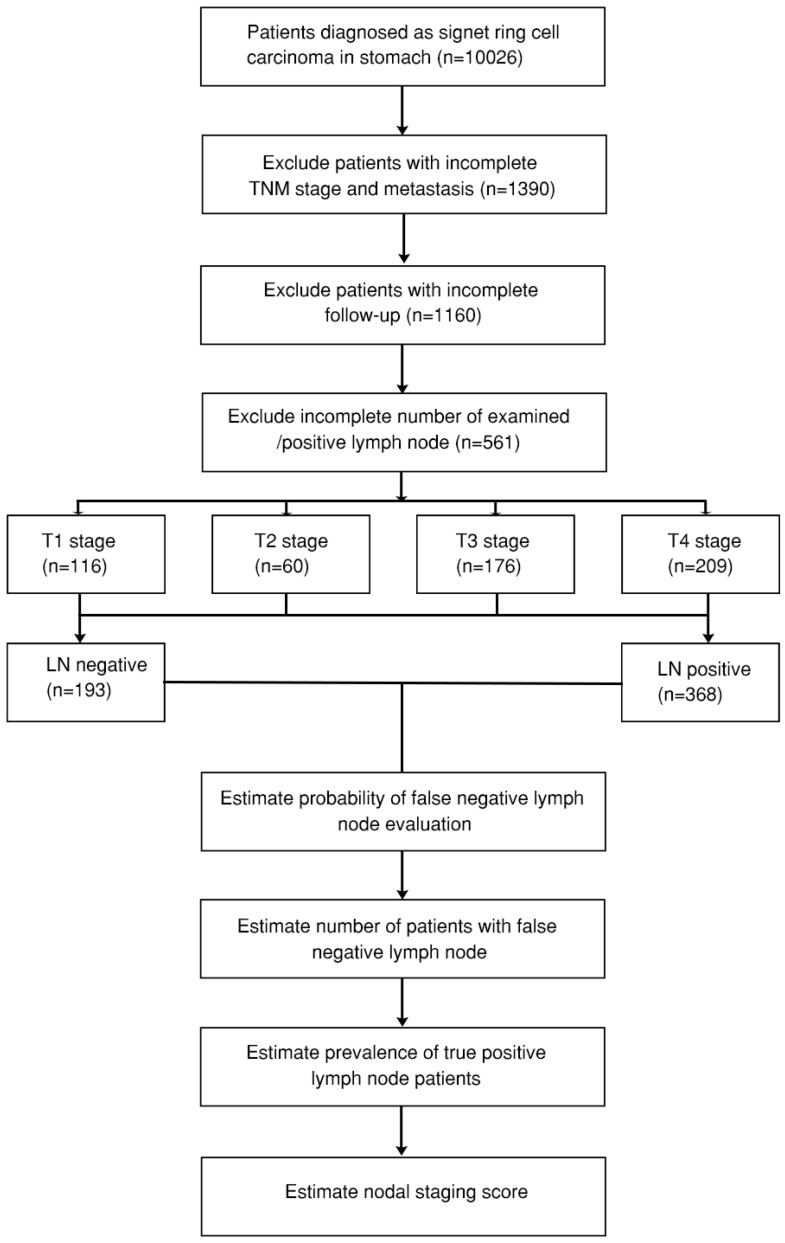
Consort diagram of gastric signet ring cell carcinoma (GSRCC) patients from the Surveillance, Epidemiology, and End Results (SEER) database. LN: lymph node; TNM stage: the American Joint Committee on Cancer (AJCC) TNM stage.

**Figure 2 diagnostics-12-02289-f002:**
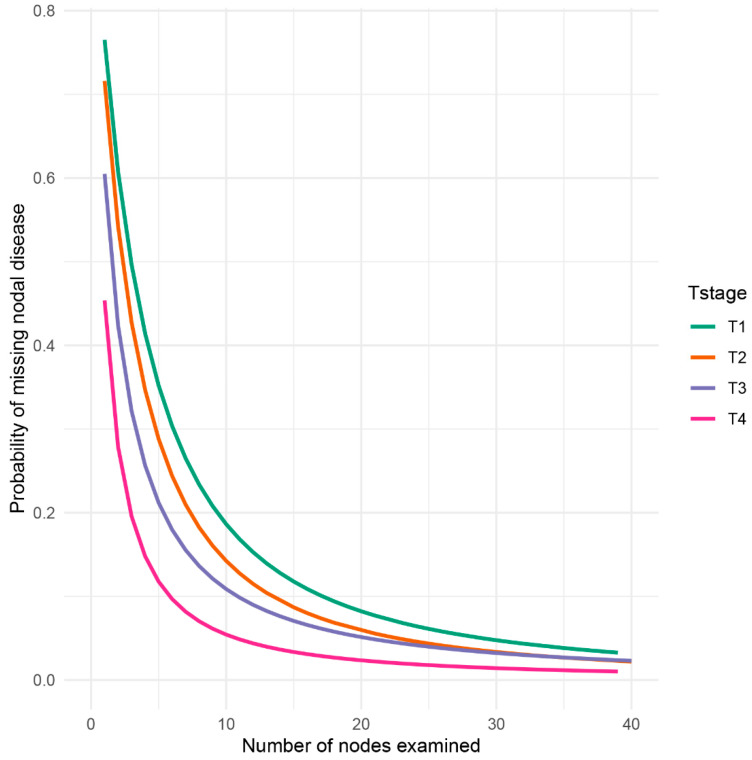
Probability of failure to identify metastatic nodes (false negative) in the pathological evaluation process. The probability was divided into four groups based on *T stage*, from T1 to T4, to examine a patient with a truly positive lymph node status.

**Figure 3 diagnostics-12-02289-f003:**
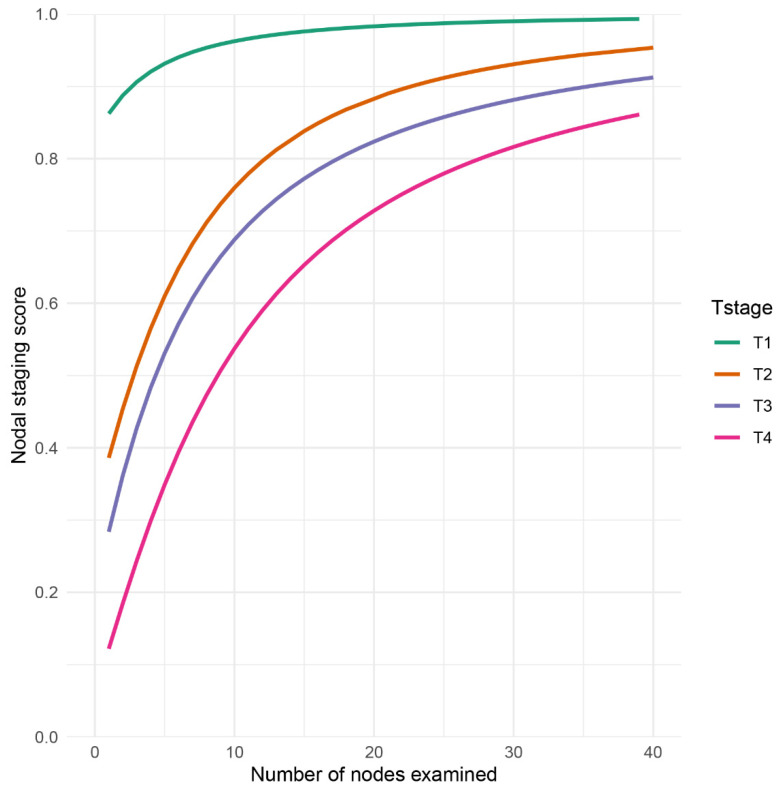
Nodal staging score (NSS) as a function of examined lymph node in GSRCC patients. NSS was displayed and stratified by *T stage*.

**Figure 4 diagnostics-12-02289-f004:**
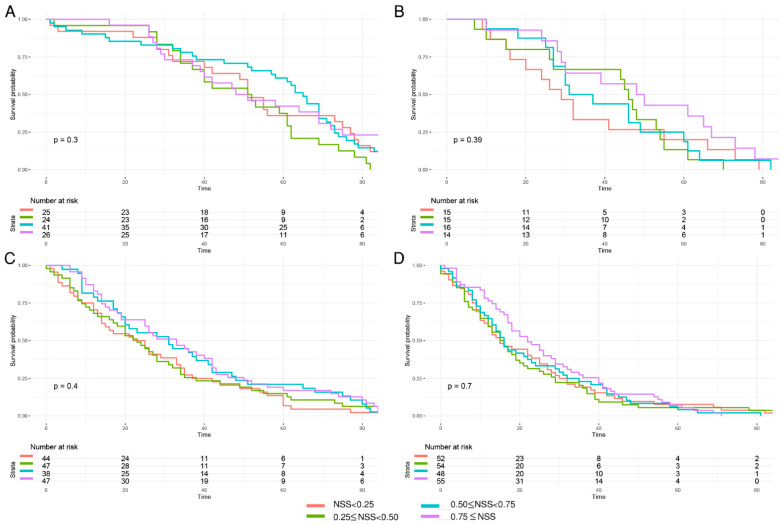
Kaplan–Meier curves of overall survival in GSRCC stratified by quantile NSS values in each T stage group. (**A**) The quantile cutoff values in T1 were 0.9625765, 0.9740264, 0.9859350; (**B**) the quantile cutoff values in T2 were 0.7540709, 0.8635734, 0.9161756; (**C**) the quantile cutoff values in T3 were 0.7227524, 0.8058663, 0.8680536; (**D**) the quantile cutoff values in T4 were 0.6129243, 0.7015987, 0.7956004.

**Table 1 diagnostics-12-02289-t001:** Characterization of included patients in this study divided by lymph node (LN) metastatic negative or positive status.

		LN Negative	LN Positive	*p* Value
*n*		193	368	
T stage (%)				
	T1	98 (50.8)	18 (4.9)	<0.001
	T2	25 (13.0)	35 (9.5)	
	T3	48 (24.9)	128 (34.8)	
	T4	22 (11.4)	187 (50.8)	
N stage (%)				
	N0	193 (100.0)	0 (0.0)	<0.001
	N1	0 (0.0)	95 (25.8)	
	N2	0 (0.0)	89 (24.2)	
	N3	0 (0.0)	184 (50.0)	
M stage (%)				
	M0	182 (94.3)	308 (83.7)	0.001
	M1	11 (5.7)	60 (16.3)	
Sex (%)				
	Male	108 (56.0)	179 (48.6)	0.119
	Female	85 (44.0)	189 (51.4)	
Age (years) (%)				
	<30	8 (4.1)	7 (1.9)	0.137
	≥70	70 (36.3)	112 (30.4)	
	30≤ <50	31 (16.1)	58 (15.8)	
	50≤ <70	84 (43.5)	191 (51.9)	
Tumor size (%)				
	<3 cm	89 (46.1)	42 (11.4)	<0.001
	3 cm≤ <6 cm	16 (8.3)	152 (41.3)	
	≥6 cm	39 (20.2)	125 (34.0)	
	unknown	49 (25.4)	49 (13.3)	
Bone metastasis (%)				
	No	193 (100.0)	364 (98.9)	0.348
	Yes	0 (0.0)	3 (0.8)	
	Unknown	0 (0.0)	1 (0.3)	
Brain metastasis (%)				
	No	193 (100.0)	366 (99.5)	0.779
	Yes	0 (0.0)	0 (0.0)	
	Unknown	0 (0.0)	2 (0.5)	
Liver metastasis (%)				
	No	192 (99.5)	367 (99.7)	0.296
	Yes	1 (0.5)	0 (0.0)	
	Unknown	0 (0.0)	1 (0.3)	
Lung metastasis (%)				
	No	193 (100.0)	366 (99.5)	0.591
	Yes	0 (0.0)	1 (0.3)	
	Unknown	0 (0.0)	1 (0.3)	
Tumor position (%)				0.007
	C16.1-Fundus of stomach	7 (3.6)	13 (3.5)	
	C16.2-Body of stomach	24 (12.4)	28 (7.6)	
	C16.3-Gastric antrum	53 (27.5)	120 (32.6)	
	C16.4-Pylorus	7 (3.6)	11 (3.0)	
	C16.5-Lesser curvature of stomach NOS	17 (8.8)	41 (11.1)	
	C16.6-Greater curvature of stomach NOS	15 (7.8)	15 (4.1)	
	C16.8-Overlapping lesion of stomach	14 (7.3)	60 (16.3)	
	C16.9-Stomach, NOS	26 (13.5)	28 (7.6)	
	C16.0-Cardia, NOS	30 (15.5)	52 (14.1)	

NOS: not otherwise specified; LN: lymph node.

**Table 2 diagnostics-12-02289-t002:** *α*_T stage_*β*_T stage_ parameters determined by fitness of *β*-binomial model. CI: confidential interval.

T Stage	α_T stage_ (95% CI)	β_T stage_ (95% CI)
T1	0.06938207 (0.03686795–0.1235056)	1.8877199 (1.2074176–4.266865)
T2	0.42578161 (0.26941777–0.7476104)	2.2180490 (1.4385130–4.397587)
T3	0.48686973 (0.38153144–0.6422990)	1.2430379 (0.9737503–1.666755)
T4	0.80543434 (0.64691993–1.0226410)	0.8710085 (0.7142595–1.084519)

**Table 3 diagnostics-12-02289-t003:** Both apparent and adjusted prevalence results from each *T stage*.

T Stage AJCC	Apparent Prevalence (%)	Adjusted Prevalence (%)
T1	0.173	0.155
T2	0.690	0.583
T3	0.807	0.727
T4	0.941	0.895

## Data Availability

The datasets supporting the conclusion of this article are included within the article. The study did not report any data.

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
