# Peer review of "Pathological Nodal Staging Score for Gastric Signet Ring Cell Carcinoma: A Clinical Tool of Adequate Nodal Staging"

_diagnostics, 2022, doi:10.3390/diagnostics12102289_

Round 1
Reviewer 1 Report
Here the authors attempted to propose a need for larger lymph node dissection in gastric cancer.
1)Why choose signet ring cell? This is not clear as many of the citations are not specific for this disease.
2) There is great variability in what many practitioners call signet ring and the proportion of those cells. How do you account for this?
3) How do you propose obtaining a higher yield? Should we be doing a full D2/D3 v a modified D2? If so, should we do more studies on this in this more aggressive subset? Or should our pathologists be looking harder for nodes?
4) How do you account for nodal differences in older v younger patients and after neoadjuvant?
Author Response
Manuscript ID: diagnostics-1879112
Title: Pathological nodal staging score for gastric signet ring cell carcinoma: a clinical tool of adequate nodal staging
Dear editors of Diagnostics,
We are very honored to hear from you and extremely appreciated for the precious revision opportunity you offered. Your efforts enable the improvement of this manuscript. All the concerns raised by the reviewers had been discussed. Modifications had been made accordingly.
Hereby, with your permission, we would like to respond to each concern from the reviewers in detail.
Reviewer A
Here the authors attempted to propose a need for larger lymph node dissection in gastric cancer.
1)Why choose signet ring cell? This is not clear as many of the citations are not specific for this disease.
Answer:Thank you for the comments. Gastric cancer is one of the intensively investigated fields in our group. We have previously published several studies focusing on multi-dimensional features of gastric cancer (1-12), wherein gastric signet ring cell carcinoma (GSRCC) is the most challenging subtype with advanced lymph node stages based on our clinical practices and research results. Citations for GSRCC have been updated as more specific references being added.
2) There is great variability in what many practitioners call signet ring and the proportion of those cells. How do you account for this?
Answer:The definition of GSRCC has always been one of the topics under disputation. In this manuscript, the definition of GSRCC have been under full discussion. Numerous history names of signet ring cell adenocarcinoma were diffuse type of gastric cancer, infiltrative type of gastric cancer, undifferentiated type of gastric cancer, high grade of gastric cancer (13-15). However, according to the latest histological classification of World Health Organization, GSRCC is commonly defined as poorly cohesive carcinoma type with prominent cytoplasmic mucin and eccentrically placed nucleus. During the data retrieving process in SEER database, only the 8490/3: Signet ring cell carcinoma from ICD-O-3 Hist/behave, malignant cases were screened for analysis.
3) How do you propose obtaining a higher yield? Should we be doing a full D2/D3 v a modified D2? If so, should we do more studies on this in this more aggressive subset? Or should our pathologists be looking harder for nodes?
Answer:Thank you for the comments. Actually, based on the conclusion of this manuscript, we highlighted the value of higher number of dissected lymph nodes. But, it is reasonably to presume that there is little room to increase the number of lymph nodes when the standard surgery procedures have been optimized. Although a more radical resection, such as D2+ or D3 for GSRCC may yield higher number of lymph nodes, which may guarantee an adequate lymph node staging, but such radical surgical treatment may increase postoperative complications as well, which may compromise the survival benefits brought by adequate lymph node staging. Meanwhile, we believe our pathologists have been endeavoring to identify potential positive lymph nodes.
Therefore, the adequate lymph node staging by NSS in this manuscript indeed proposes a predictive tool for both diagnosis and subsequent treatment. For example, if standard operation for a T4 GSRCC case yielded a total of 35 negative lymph nodes, it is quantificationally sure that the NSS will be >0.8 with considerably less possibility to have false negative findings. Meanwhile, it also actively feedbacks the surgery performance.
4) How do you account for nodal differences in older v younger patients and after neoadjuvant?
Answer:As far as we checked, the neoadjuvant therapy information is probably not available among most of GSRCC patients in the SEER database. According to our previously published studies, elderly gastric cancer was defined as age >=70 years old while the young gastric cancer was defined as age<45 years old (2,12). The lymph node staging, as well as other variables were demonstrated among young GSRCC, elderly GSRCC and the rest groups (Table S1). In fact, no significant lymph node staging was found among each group.
A thorough linguistic review would be required.
Answer:Thank you for the comments. The manuscript has been under fully linguistic revision to ensure improved readability
Best
Yousheng Li, M.D., Ph.D.,
Department of General Surgery,
Shanghai Ninth People's Hospital, Shanghai Jiao Tong University School of Medicine, Shanghai, People's Republic of China; Shanghai 200025, P.R.China;
E-mail: guttx@hotmail.com.
Reference
- Zhang Y, Yu C. Bibliometric Evaluation of Publications (2000-2020) on the Prognosis of Gastric Cancer. Inquiry. 2021 Jan-Dec;58:469580211056015. doi: 10.1177/00469580211056015.
- Zhang Y, Yu C. Development and validation of a Surveillance, Epidemiology, and End Results (SEER)-based prognostic nomogram for predicting survival in elderly patients with gastric cancer after surgery. J Gastrointest Oncol. 2021 Apr;12(2):278-296. doi: 10.21037/jgo-20-536.
- Wang Q, Yu C. Letter to editor: development and internal validation of a diagnostic score for gastric linitis plastic. Gastric Cancer. 2021 Sep;24(5):1167-1168. doi: 10.1007/s10120-021-01192-7.
- Yu C. Comment on: "Hepatoid adenocarcinoma of the stomach: a unique subgroup with distinct clinicopathological and molecular features. Gastric Cancer, 2019: 1-10" by Wang et al. Gastric Cancer. 2019 Nov;22(6):1312. doi: 10.1007/s10120-019-00996-y.
- Yu C, Helwig EJ. Artificial intelligence in gastric cancer: a translational narrative review. Ann Transl Med. 2021 Feb;9(3):269. doi: 10.21037/atm-20-6337.
- Zhang Y, Yu C. Distinct expression and prognostic values of the replication protein A family in gastric cancer. Oncol Lett. 2020 Mar;19(3):1831-1841. doi: 10.3892/ol.2020.11253.
- Yu C, Hao X, Zhang S, Hu W, Li J, Sun J, Zheng M. Characterization of the prognostic values of the NDRG family in gastric cancer. Therap Adv Gastroenterol. 2019 Jul 21;12:1756284819858507. doi: 10.1177/1756284819858507.
- Yu C, Zhang Y. Characterization of the prognostic values of CXCR family in gastric cancer. Cytokine. 2019 Nov;123:154785. doi: 10.1016/j.cyto.2019.154785.
- Yu C, Xue P, Zhang L, Pan R, Cai Z, He Z, Sun J, Zheng M. Prediction of key genes and pathways involved in trastuzumab-resistant gastric cancer. World J Surg Oncol. 2018 Aug 22;16(1):174. doi: 10.1186/s12957-018-1475-6.
- Yu C, Chen J, Ma J, Zang L, Dong F, Sun J, Zheng M. Identification of Key Genes and Signaling Pathways Associated with the Progression of Gastric Cancer. Pathol Oncol Res. 2020 Jul;26(3):1903-1919. doi: 10.1007/s12253-019-00781-3.
11.Xie D, Yu C, Liu L, Osaiweran H, Gao C, Hu J, Gong J. Short-term outcomes of laparoscopic D2 lymphadenectomy with complete mesogastrium excision for advanced gastric cancer. Surg Endosc. 2016 Nov;30(11):5138-5139. doi: 10.1007/s00464-016-4847-4.
- Yu C, Zhang Y. Development and validation of prognostic nomogram for young patients with gastric cancer. Ann Transl Med. 2019 Nov;7(22):641.
- LAUREN P. THE TWO HISTOLOGICAL MAIN TYPES OF GASTRIC CARCINOMA: DIFFUSE AND SO-CALLED INTESTINAL-TYPE CARCINOMA. AN ATTEMPT AT A HISTO-CLINICAL CLASSIFICATION. Acta Pathol Microbiol Scand. 1965;64:31-49. doi: 10.1111/apm.1965.64.1.31. PMID: 14320675.
- Ming SC. Gastric carcinoma. A pathobiological classification. Cancer. 1977 Jun;39(6):2475-85. doi: 10.1002/1097-0142(197706)39:6<2475::aid-cncr2820390626>3.0.co;2-l. PMID: 872047.
- Patel MI, Rhoads KF, Ma Y, Ford JM, Visser BC, Kunz PL, Fisher GA, Chang DT, Koong A, Norton JA, Poultsides GA. Seventh edition (2010) of the AJCC/UICC staging system for gastric adenocarcinoma: is there room for improvement? Ann Surg Oncol. 2013 May;20(5):1631-8. doi: 10.1245/s10434-012-2724-5. Epub 2012 Nov 13. PMID: 23149854.
Reviewer 2 Report
I thank the Editor for submitting this paper to me for re-viewing.
The topic is very interesting but there are important biases.
First of all, the therapeutic choice for pT1 gastric tumors involves less surgical aggression, and therefore it is normal that there are fewer lymph nodes in this case.
Second: the different risk classes should be related to other histotypes and not to 'gastric adenocarcinoma' in general.
Third: the number of lymph nodes should be correlated with the stages (pT pN pM) and not only with the pT value (which does not constitute, in itself, staging).
The text is often unclear, the statements unrelated to the previous and subsequent paragraphs.
Finally, a thorough linguistic review would be required.
Author Response
Manuscript ID: diagnostics-1879112
Title: Pathological nodal staging score for gastric signet ring cell carcinoma: a clinical tool of adequate nodal staging
Dear editors of Diagnostics,
We are very honored to hear from you and extremely appreciated for the precious revision opportunity you offered. Your efforts enable the improvement of this manuscript. All the concerns raised by the reviewers had been discussed. Modifications had been made accordingly.
Hereby, with your permission, we would like to respond to each concern from the reviewers in detail.
Reviewer B
I thank the Editor for submitting this paper to me for re-viewing. The topic is very interesting but there are important biases. First of all, the therapeutic choice for pT1 gastric tumors involves less surgical aggression, and therefore it is normal that there are fewer lymph nodes in this case.
Answer:Thank you for the comments. Your suggestions are valuable to the improvement of this manuscript. Indeed, pT1 early gastric cancer usually involves less radical surgery than advanced gastric cancer, which is commonly related to fewer yielding of lymph nodes. In fact, the probability of missing potential metastatic lymph node was calculated and determined by the examined and resected lymph nodes of GSRCC patients retrieved from SEER database. However, only the surgery status was revealed, with the exact surgical management (D1/D1+/D2/D2+) remains unavailable. Therefore, it is indeed that, in this manuscript, fewer dissected lymph nodes and less positive lymph nodes maybe found in early T stage instead of advanced T stage (Figure S1, S2). You are very insightful.
Second: the different risk classes should be related to other histotypes and not to 'gastric adenocarcinoma' in general.
Answer:Thank you for the comments. It is absolutely right that the different risk classes should be analyzed based on all types of histologic types of gastric cancer, instead of one type only. In fact, difference histologic types of gastric cancer may be featured by distinct risk variables. We deeply appreciate your suggestion on this, and are willing to extend the research of NSS into other histologic types of gastric cancer.
Third: the number of lymph nodes should be correlated with the stages (pT pN pM) and not only with the pT value (which does not constitute, in itself, staging).
Answer:Thank you for the suggestions. Based on previous similar NSS studies and original algorithms developed by Gönen et al. in 2009 (1-3), one of the main bedrock theories for NSS is to compute the prevalence of potential nodal disease as a function of T stage and the probability of missing a metastatic node as a function of the number of examined nodes. Gönen et al. indicated that the nodal prevalence rather inherent to the disease, instead of the pathological detection, whereas the probability is inherent to the pathological detection but independent to the T stage. Therefore, NSS is generated based on the function of both T stage and the number of examined nodes. In fact, as the number of lymph nodes, both examined and positive data, has been incorporated into the NSS algorithm processing, it indicates that the pN may not be precise, which is why NSS is developed. In addition, we believe pM is an insightful option for developing NSS algorithm. However, given the fact that most of distant metastasis patients may not be able to receive a standard operation, therefore, it may not be feasible to yield sufficient lymph nodes for NSS assessment. Nonetheless, increasing evidences have supported that surgical treatment for stage IV gastric cancer may also be beneficial to prognosis. Thus, we will value your comments and hope to add the idea into future study with more available M1 patients.
The text is often unclear, the statements unrelated to the previous and subsequent paragraphs. Finally, a thorough linguistic review would be required.
Answer:Thank you for the comments. The manuscript has been under fully linguistic revision to ensure improved readability
Best
Yousheng Li, M.D., Ph.D.,
Department of General Surgery,
Shanghai Ninth People's Hospital, Shanghai Jiao Tong University School of Medicine, Shanghai, People's Republic of China; Shanghai 200025, P.R.China;
E-mail: guttx@hotmail.com.
Reference
- Wu Z, Qin G, Zhao N, Jia H, Zheng X. Assessing the adequacy of lymph node yield for different tumor stages of colon cancer by nodal staging scores. BMC Cancer. 2017;17(1):498.
- Gönen M, Schrag D, Weiser MR. Nodal staging score: a tool to assess adequate staging of node-negative colon cancer. J Clin Oncol. 2009 Dec 20;27(36):6166-71.
- Robinson TJ, Thomas S, Dinan MA, Roman S, Sosa JA, Hyslop T. How Many Lymph Nodes Are Enough? Assessing the Adequacy of Lymph Node Yield for Papillary Thyroid Cancer. J Clin Oncol. 2016 Oct 1;34(28):3434-9.
Round 2
Reviewer 2 Report
The paper has been improved